# Snail Slime Extracted by a Cruelty Free Method Preserves Viability and Controls Inflammation Occurrence: A Focus on Fibroblasts

**DOI:** 10.3390/molecules28031222

**Published:** 2023-01-26

**Authors:** Alessia Ricci, Marialucia Gallorini, Nadine Feghali, Simone Sampò, Amelia Cataldi, Susi Zara

**Affiliations:** 1Department of Pharmacy, University “G. d’Annunzio” Chieti-Pescara, 66100 Chieti, Italy; 2International Institution of Heliciculture of Cherasco-Lumacheria Italiana s.r.l., 12062 Cherasco, Italy

**Keywords:** snail slime, skin, fibroblasts, inflammation, keratinocytes, macrophages, cell survival, angiogenesis, collagen

## Abstract

Snail slime (SS) is a viscous secretion obtained from different snail species. SS composition is variable according to factors such as the extraction method. Even if several papers have been published regarding this topic, the molecular mechanisms at the base of SS biological effects remain unexplored. Thus, the aim of this study is to evaluate the capability of SS, extracted with the cruelty-free Muller method, to promote viability and angiogenesis processes and, in parallel, to counteract inflammation occurrence on skin cell populations. SS was administered to keratinocytes, macrophages and fibroblasts, then cell viability, through 3-(4,5-dimethylthiazol-2-yl)-2,5-diphenyltetrazolium bromide (MTT) test, cytotoxicity by lactate dehydrogenase (LDH) assay, morphology by haematoxylin-eosin staining, gene and protein expression through real-time polymerase chain reaction (PCR) and Western blot, cell cycle phases by flow cytometry, and collagen secretion using an enzyme-linked immunosorbent assay (ELISA) test, were measured. Our results evidence SS capability to promote fibroblast viability and to trigger recovery mechanisms by activating the Erk protein. Moreover, an appreciable anti-inflammatory effect due to the significant reduction in cyclooxygenase-2 expression, and a positive modulation of new blood vessel formation demonstrated by increased Angiopoietin 1 gene expression and a higher matrix deposition (evidenced by the augmented amount of released collagen I) can be identified. This evidence led us to assume that the Muller method extracted-SS represents a valuable and promising natural product suitable for cosmetic and skin care formulations.

## 1. Introduction

Snail slime (SS) is defined as a viscous and liquid secretion with a very low pH (approximately three), obtained from different gastropod species, such as *Helix* species. This hydrogel-like mucus is secreted by different glands: for example, *Helix pomatia* specie releases SS from five different gland types, three located on the dorsal side and two on the ventral side of the foot sole. SS secretion is essential for snails, allowing their adhesion and locomotion on different types of surfaces, preventing excessive animal drying, and is necessary for feeding, reproduction, and representing a defensive substance against predators [1]. SS useful properties are well known; indeed, it has been reported that our ancestors treated skin injuries such as burns, abscesses and other wounds using snail secretion which has also been found to be part of dermatologic preparations [2]. Today, SS is still used in several applications for its healing properties. As an example, *Helix pomatia* extract can be found in different preparations in particular syrups, which exert a broncho-relaxant effect increasing prostaglandin E2 in the respiratory system [3]. Moreover, SS has been described as an essential element to enrich gelatin-based films designed for skin drug delivery systems, thanks to the optimal adhesion and film flexibility [4]. Recently, SS has gained popularity for its cosmetic and skincare properties, such as antiaging, skin regeneration, acne control, hydrating effects, and many others, due to several antioxidant, anti-inflammatory, and pro-regenerating components including allantoin, glycolic acid, elastin, collagen, vitamins, and mucin-like glycoprotein complexes, which exert sun-protecting, exfoliating, and many other beneficial effects [5]. The composition of SS depends on different factors, such as the snail species, feeding and behavior of the animals and, most importantly, the extraction method [4].

The human skin is composed of three layers represented by epidermidis, derma, and hypodermis. The most represented cell populations are keratinocytes, macrophages, and fibroblasts. Gubitosa and colleagues tested snail mucus conjugated with gold nanoparticles (AuNP-SS), finding wound-healing effects on keratinocytes and anti-inflammatory properties on macrophages [6]. In a second study, they also found in AuNP-SS sunscreen and tyrosinase inhibition abilities [7], confirming the promising properties of SS conjugated with gold nanoparticles. Liposomes have been also used as vehicle for snail mucus and the properties of this innovative formulation were studied on fibroblasts, highlighting again a wound-healing ability extremely useful for skin recovery [8]. Finally, Perpelek’s group focused its attention on the capability of slime and mucus extracts loaded in a chitosan matrix, to promote bone and cartilage regeneration [9].

However, despite the large number of commercially available products containing SS and the growing interest in it, most of the published papers regarding this topic focus on SS biological effects without studying the underlying molecular mechanisms in depth.

This study is designed to test SS on the three most abundant cell populations of the skin with the aim of demonstrating the beneficial effect of SS in terms of cell viability, proliferation, control of inflammatory events and angiogenesis promotion. SS is extracted with the innovative and cruelty-free Muller method, performed using Muller One instruments, which implies a continuous cyclic stimulation of snail secretion by rinsing snails with stimulating solutions of citric acid.

## 2. Results

### 2.1. Evaluation of Keratinocytes Viability

Keratinocytes were the first cell population in contact with formulations for topic administrations. SS dilutions were firstly tested by means of 3-(4,5-dimethylthiazol-2-yl)-2,5-diphenyltetrazolium bromide (MTT) on HaCat keratinocytes cell lines for 24, 48, 72 h, and 6 days. Dilutions of 1:20, 1:40, 1:60, and 1:80 within the culture medium were chosen (Figure 1).

SS was administered at 1:20 and an appreciable and significant increase in viable cells was detected after treatment with 1:60 and 1:80 dilutions, compared with the control (CTRL1). After 48 h of treatment, again, the SS dilution of 1:20 led to a statistically significant reduction in cell viability compared with the control. After 72 h and 6 days, SS dilutions of 1:20 and 1:40 provoked a significant decrease in cell viability, with a major extent for dilution 1:20.

### 2.2. Analysis of Macrophage Viability and Activation

Human macrophages were firstly exposed to decreasing concentrations of SS (1:20, 1:40, 1:60, and 1:80) to evaluate its biocompatibility and the activation rate on immune cells as very sensitive ones (Figure 2A).

After 24 h of exposure, macrophages treated with the highest concentration of SS showed a dramatic decrease in cell metabolic activity (37.4%) compared with the control sample, while the 1:40 dilution doubled metabolically active cells (74%), disclosing a dose-dependent viability trend. On the contrary, lower dilutions of SS did not affect macrophage viability. The same pattern was detected after 48 h of treatment.

Secondly, macrophages were stimulated with a low concentration of lipopolysaccharide (LPS) (0.5 µg/mL) to establish an inflamed environment, and the SS was administered in parallel at 1:40, 1:60, and 1:80 dilutions. As expected, the LPS stimulation activated macrophages after 24 h, and the cell percentage of metabolically active cells was assessed at 213.7% with respect to the control. The SS administered in parallel seems capable of counteracting this activation, and percentages of viability were comparable with the control. After 48 h, the co-treatment LPS/SS significantly decreased the percentages of metabolically active cells compared with the control and LPS, mainly at 1:40 and 1:60 dilutions (Figure 2B).

### 2.3. Evaluation of Human Gingival Fibroblasts (HGFs) Viability and Cytotoxicity

Secondly, considering that after passing the epidermidis, dermatological products will be in contact with the connective tissue cells, SS was tested on HGFs by measuring cell viability and cytotoxicity. HGF viability was measured after exposure to SS for 24, 48, and 72 h. Dilutions of SS of 1:40, 1:60, and 1:80 were examined; SS dilution of 1:20, responsible for a high rate of dead keratinocytes, was eliminated from the screening panel. After 24 h of treatment there were no significant differences in cell response, whereas, after 48 and 72 h, a significant reduction in cell viability level is recorded. When SS is administered at 1:40 and 1:60 dilutions, the viable cells percentage is still maintained over 70% (Figure 3A). Then, the cytotoxicity level, through the measurement of lactate dehydrogenase (LDH) release within the culture medium, was evaluated after exposure to the minimal (1:80) and to the maximal (1:40) concentrations of SS for 24 and 48 h. After 24 h of treatment, no statistically significant differences were recorded among the tested samples and after 48 h a statistically significant reduction in cytotoxicity level was measured in HGFs exposed to both 1:40 and 1:80 SS dilutions compared with untreated HGFs (CTRL1) (Figure 3B).

### 2.4. Morphological Evaluation

HGFs exposed to SS (CTRL1, 1:40 and 1:80) and pre-treated with H_2_O_2_ 100 μM and then exposed to SS (CTRL2, H_2_O_2_ + 1:40 and H_2_O_2_ + 1:80) were stained by means of haematoxylin/eosin after 24 h of treatment (Figure 4).

CTRL1, 1:40, and 1:80 all highlight viable cells with an elongated shape, totally adherent to the plate surface with a high rate of cell confluence. No dead cells are detected. HGFs exposed to H_2_O_2_ 100 μM (CTRL2) show several cells close to the detachment (black arrows) along with a lower cell confluence, while in samples H_2_O_2_ + 1:40 and H_2_O_2_ + 1:80 the situation appears reversed with large areas of elongated and viable cells.

### 2.5. Cell Cycle Analysis

A cell cycle analysis was performed after 6 and 24 h of treatment by flow cytometry in order to evaluate the HGF distribution in the cell cycle phases. Untreated cells (CTRL1) show a physiological distribution of cell cycle phases and they seem actively proliferative at both time points (G1 phase = 75% (6 h)–83% (24 h); S phase = 12% (6 h)–5% (24 h); G2 phase = 12% (6 h)–11% (24 h)). After 6 h of treatment with SS 1:40 there is a statistically significant increase in cell percentage in the G1 phase (78.5%) with a slight but significant decrease in cell percentage in the G2 phase (10%) compared to CTRL1. On the contrary, the administration of SS 1:80 significantly decreases the percentage of cells in the G1 phase (G1 = 73.5%) compared to CTRL1. After 24 h, the percentage of cells in the G1 phase decreases in a statistically significant manner with SS 1:40 (75.5%) and 1:80 (79.9%) dilution treatments compared to untreated cells, while there is a higher and significant increase in cell percentage in the S phase (1:40 = 11.6%; 1:80 = 7.9%) (Figure 5A).

The analysis of the cell cycle was also performed on HGFs after 3 h of pre-treatment with H_2_O_2_ 100 µM followed by 6 h and 24 h of treatment with 1:40 and 1:80 SS dilutions. The cell cycle analysis of the control (CTRL2) highlights the suffering profile of the cells with a slowdown in proliferation at both time points (G1 phase = 47% (6 h)–57% (24 h); S phase = 29% (6 h)–17.5% (24 h); G2 phase =23% (6 h)–25.5% (24 h)). After 6 h, H_2_O_2_ + 1:80 treatment induces a significant increase in the percentage in G1 phase cells (49%) and a significant reduction in G2 phase cells percentage (21%), compared to the CTRL2. After 24 h, H_2_O_2_ + 1:40 increases the percentage of S phase cells in a statistically significant manner (20.6%) compared to CTRL2, and, in parallel, reduces the percentage of G2 phase cells (23%). The administration of H_2_O_2_ + 1:80 significantly reduces the percentage of cells in G1 phase (54.5%) (Figure 5B).

### 2.6. Analysis of Erk Activation

The activation of the Erk survival protein was then analyzed by Western blot after 6 and 24 h of treatment.

In HGFs exposed to SS at 1:40 and 1:80 dilutions, a statistically significant increase in pErk/Erk ratio is recorded compared to the control after 6 h of treatment, with a higher statistical significance in the presence of 1:40 SS; conversely, no significant differences are recorded after 24 h of treatment (Figure 6A,B). The same trend is detected in HGFs exposed to H_2_O_2_ 100 μM and then treated with SS (Figure 6C,D).

### 2.7. Analysis of the Inflammation Occurrence

In order to estimate the occurrence of inflammatory events, the expression of the inducible protein cyclooxygenase-2 (COX-2) was measured through Western blot after 6 h of treatment. In HGFs exposed to SS at 1:40 and 1:80 dilutions, a significant reduction in COX-2 expression is recorded compared to CTRL1, with a major extent for 1:40 dilution, and the same trend is recorded for HGFs pre-treated with H_2_O_2_ 100 μM and then exposed to SS at 1:40 and 1:80 dilutions (Figure 7).

### 2.8. Evaluation of the Angiogenesis Induction

The angiogenic process occurrence was evaluated by measuring the Angiopoietin 1 (ANGPT1) and Angiopoietin 2 (ANGPT2) gene expression after 6 and 24 h of treatment. The gene expression of ANGPT1 and ANGPT2 in all the experimental samples was compared to ANGPT1 gene expression of CTRL1, chosen as calibrator sample (calibrator sample defined as 1). After 6 h of treatment a statistically significant reduction in ANGPT1 gene expression was recorded in CTRL2 compared to CTRL1, 1:40, and H_2_O_2_ + 1:40 (Figure 8A). After 24 h of treatment, a significant augmentation of ANGPT1 gene expression is disclosed in H_2_O_2_ + 1:80 compared to CTRL1; furthermore, a statistically significant reduction can be evidenced in CTRL2 compared to CTRL1, 1:40, and H_2_O_2_ + 1:80 (Figure 8B).

ANGPT2 mRNA levels were also measured finding the mRNA expression only after 6 h of treatment in CTRL1 and in HGFs exposed to SS at 1:40 dilution (Figure 9). The ANGPT2 gene expression in 1:40 appears significantly reduced compared to ANGPT2 gene expression in CTRL1. All other tested samples and experimental times revealed no gene expression for ANGPT2.

### 2.9. Collagen Secretion

Lastly, the HGF capability to synthetize collagen I and secrete it within the culture medium was measured through an ELISA assay after 24 and 48 h of treatment. After 24 h of treatment, HGFs cultured with SS at both 1:40 and 1:80 dilutions evidence a higher and significant collagen I secretion compared to the control (Figure 10, left histogram). The same trend with increased secretion levels is recorded after 48 h of treatment. Conversely, when HGFs are pre-treated with H_2_O_2_ 100 μM and then exposed to SS, a statistically significant increase in collagen I secretion in H_2_O_2_ + 1:40 and H_2_O_2_ + 1:80 compared to the control is recordable only after 48 h of treatment (Figure 10, right histogram).

## 3. Discussion

SS has been used for centuries to treat dermatological wounds, burns, and abscesses. Today, preparations with SS are also used to treat numerous skin disorders [2]. To date, few studies focused on the molecular pathways underlying the biological effects of SS on skin cell populations. The majority of the available papers, in fact, describe the effect of SS after conjugation with vehicles, such as gold nanoparticles or liposomes [6,7,8]. Thus, in this paper SS was tested on the three most represented cell populations of the skin deepening the possible signaling pathway involved in its mechanism of action.

The epidermidis was the first skin layer encountered in a topic application [10] and SS was tested on human keratinocytes. This preliminary evaluation shed light on the fact that SS lower concentrations are well tolerated by keratinocytes while the highest concentration (1:20) needs to be excluded from further investigations as a possible keratinolytic effect was markedly evidenced. This could be related to the large amount of glycolic acid, which has a well-known keratinolytic effect in SS composition. Several medical and cosmetic preparations, aiming at obtaining an exfoliating effect to treat acne, melasma, and post-inflammatory hyperpigmentation, include glycolic acid in their formulations [11,12].

The SS effect was then tested on primary human fibroblasts highly represented within the dermis layer. Our results are in accordance with the study of Alogna and colleagues [8], which had already demonstrated that fibroblasts are well known to tolerate snail mucus administration, resulting in an appreciable HGF viability level after SS administration. This result is further supported by the LDH release measurement which markedly underlines the notable SS capability to counteract the cytotoxicity condition, probably triggering a cell protection mechanism.

The skin is constantly and intensely exposed to environmental factors such as mechanical trauma, irradiation, chemicals, etc., and are often responsible for inflammatory events. They modify mechanosensitive pathways which, in turn, impacts skin homeostasis. At the same time, in some patients intrinsic factors including autoimmune responses, mutations, and others, can be responsible for inflammatory skin disorders [13]. For instance, the two most diffuse inflammatory skin conditions mediated by autoimmune response are psoriasis and atopic dermatitis, both involving T-cells [14]. In the first case, T-helper 17 is implicated in the pathogenesis of psoriasis, and interleukins IL-17, IL-22, and IL-23 are the main cytokines involved [15]. In atopic dermatitis, or eczema, the main role is played by T-helper 2 cells and an upregulation of IL-4 and IL-13 is found, in addition to interferon (IFN)-γ, IL-17, and IL-22 in chronic eczema [16]. In both conditions, a signaling cascade is triggered, starting with the activation of the immune system, with immune cell infiltration into the epidermis and dermis, followed by pro-inflammatory cytokines released by fibroblasts, keratinocytes, and the same immune cells. Simultaneously, there is a hyperproliferation and improper differentiation of keratinocytes and both keratinocytes and fibroblasts shift to the wound-healing phenotype [17]. As SS preparations are not considered pharmacologic preparations, their use in pathological conditions such as psoriasis or dermatitis is not evaluable. However, the role of SS in the prevention or treatment of inducible inflammatory conditions could be an intriguing topic. The ability of SS to counteract inflammation events has been also investigated by exposing fibroblasts to H_2_O_2_ in order to generate an inflamed environment, and then treating them with SS. SS showed a prompt capacity to induce a compensatory response thus improving fibroblasts viability. This hypothesis appears supported by SS capability to definitely decrease the expression of pro-inflammatory COX-2 at very early times of exposure. In fact, in several cell types, among which fibroblasts are included [18], prostaglandins release and the maintenance of a phlogosis condition are triggered by COX-2 increased levels. Furthermore, in our experimental model SS showed a marked capability to effectively counteract COX-2 basal expression in HGFs not exposed to pro-inflammatory stimuli, thus enlarging the range of its beneficial effects. Unfortunately, SS capability to counteract the recruitment of pro-inflammatory molecules appears to flatten over 6 h of treatment, probably suggesting the necessity to refresh the SS treatment to keep an appreciable anti-inflammatory effect.

Tissue macrophages are the most abundant resident immune cell type of the skin, participating in homeostasis and healing processes. Under homeostatic conditions, their primary role is to scavenge cellular and macromolecular degradation intermediates while, when stimulated by inflammatory or pathological stimuli along with microorganisms in the dermis, they become activated [19]. A major pathway in host defense is the activation of the transmembrane protein toll-like receptor 4 (TLR4) in innate immune cells. It has been reported that a chronic TLR4 stimulation in skin induces inflammation mediated by macrophage activation and transforming growth factor beta (TGFβ) signature gene expression, leading to fibrosis [20]. One of the most widely used and resourceful in vitro models of inflammation is the stimulation and activation of TLR4 in cultured monocytes/macrophages by gram-negative products or LPS [21]. In this light, establishing an inflamed environment by stimulating macrophages with LPS is a valuable strategy to study activation of these cells in our experimental model, as demonstrated by the dramatic raise of cell metabolic activity under LPS stimulation. Indeed, under LPS-induced inflammatory conditions, the sustained metabolic adaptations are functional to support macrophage activities as well as to support their polarization in specific contexts [22].

In our experimental model, lower dilutions of SS do not affect macrophage viability and the same concentrations are capable of counteracting the LPS-induced cell metabolic activity and macrophage activation. Data reported disclose a role for SS in immunomodulation and lay the grounds for further in-depth investigations on the anti-inflammatory effects of SS on the immune system.

To counteract skin aging or damage aiming at obtaining cell recovery, two crucial steps are required: the proliferation step, including phenomena such as cell renewal, epithelialization, and angiogenesis, and the maturation step, involving collagen deposition [23]. Cell proliferation is regulated by molecular cascades driving the correct transition in the cell cycle phases to gain cell turnover [24,25]. In our experimental model, cell cycle analysis confirms the previous point highlighting the proliferative potential of SS-treated HGFs: it appears, in fact, that SS is able to promote the G1–S transition, resulting in a good cell cycle turnover. The morphologic analysis also confirmed a high viability and the lack of morphological modifications for HGFs treated with SS. The promotion of survival events, driven by SS administration, is demonstrated by the analysis of Erk activation. Erk is an essential serine–threonine protein kinase positively regulating cell proliferation and survival events [26]. It appears considerably phosphorylated promoting the activation of cell cycle machinery after a SS short-time exposure reaching a plateau after longer exposure, thus underlining again the need for SS renewal within the culture medium. Conversely, the morphological analysis performed on H_2_O_2_-exposed HGFs underlines cell viability affected and several cells, even if still viable, close to the detachment. Interestingly SS administration completely reverses the effect of H_2_O_2_, and a morphological recovery appears visible. This aspect is also confirmed by cell cycle analysis and by Erk activation. Both parameters disclose a positive regulation of cell proliferation and the triggering of survival mechanisms.

In the dermis layer a crucial role is played by the large blood vessels for the delivery of nutrients to the cells, and new blood vessel formation is an essential event after inflammation or injuries. Fibroblasts, when appropriately stimulated, can release into their environment factors which are able to stimulate endothelial cells during the angiogenic process [27]. Our results demonstrated that SS administration indirectly supports the recruitment of molecular pathways leading to new blood vessel formation. ANGPT1 gene expression, which works as a protective growth factor both in physiological and pathological angiogenesis promoting vascular phenotypes [28], is appreciably increased when SS is administered after having triggered an inflammatory stimulus. In parallel, the gene expression of the ANGPT2 anti-angiogenic factor, which acts as ANGPT1 activity antagonist on tie-2 receptor, appears reduced after SS treatment, thus further supporting the positive modulation of the pro-angiogenic signaling.

In addition, our experimental model revealed that in cells pre-treated with H_2_O_2_, the SS pro-angiogenic effect, exerted by increasing the ANGPT1 gene expression, is revealed in advance compared to the standard model, thus enforcing the hypothesis that the beneficial effects of SS in the cell recovery process is strengthened in cells exposed to an inflamed environment.

Lastly, during the maturation step, cells are engaged in the secretion of molecules involved in the extracellular matrix remodeling, such as collagen type I, which represents the most abundant protein synthetized and secreted by fibroblasts [29]. As collagen I secretion increased after SS administration, both in a standard and in an inflamed-environment condition, the authors assumed that a wound healing process could be definitely facilitated by SS administration. The longer time required to record an appreciable increase in collagen I secretion in the inflamed-environment condition compared to the standard one could be related to the need for stressed HGFs to primarily be involved in viability restoring and survival proteins synthesis.

## 4. Materials and Methods

### 4.1. SS Extraction

The extraction procedure aimed at obtaining a raw material with very high chemical and organoleptic characteristics in the total well-being of the mollusk. Before starting extraction, snails of the species *Helix aspersa* were harvested as adult live snails (12–15 months), then purged leading to expulsion of excrements and cleaned drying the humidity for a minimum of 2 days. Dead snails and animals with broken shells were eliminated and the snails were carefully washed with running water at a controlled temperature avoiding thermal and traumatic shocks, and the operation was repeated at least 3 times.

Snails were placed inside the Muller extractor machine (Lumacheria Italiana srl, Cherasco, Italy).

The extraction program was set in the software installed in the machine. The procedure consists of 2 phases:

-Sanitizing/awakening of the snails: the animals were subjected to a delicate shower of osmotic water, or water mixed with ozone. This phase aimed at removing most of the bacterial load present on animals thus drastically reducing the levels of degenerative microorganisms. Afterwards, a phase of revitalization and awakening of the animals was carried out by spraying them with ozonized water for 30 min intermittently.

-Extraction of the slime: the snails were sprayed with an acid-stimulating solution taken using an electric pump (>50%), citric acid (5–10%), potassium sorbate (0.1–1%), and sodium benzoate (0.1–1%) for 30 min intermittently. The stimulating solution provided a significant amount of slime without creating any stress and damage to the snail as well as preserving the final extracted product.

A complete extraction cycle required 1 h, and afterwards the slime collected in the appropriate containers was transferred to a different container after an adequate microfiltration with the use of the filtering apparatus made up of 2 filters, one of 100 µm and the next of 50 µm, useful for the purification of any residues.

### 4.2. SS Feasibility and Stability Evaluation

After microfiltration, the slime was stabilized with the addition of natural preservatives (potassium sorbate and sodium benzoate in the amount of 1 g per kg) in order to ensure a shelf life of 18 months at room temperature. If the slime was not stabilized, it can be stored in a cold room at a temperature of +4/+10 °C for a maximum period of 4 months.

In order to achieve a standardized and stable product, every batch of extracted and microfiltrated SS was subjected to laboratory analyses aiming at controlling the organoleptic, chemical-physical and microbiological characteristics, and the presence of heavy metals (Appendix A).

The microfiltrated and stabilized SS was extracted with the Muller machine at the Lumacheria Italiana srl company (Cherasco, Italy), certified COSMOS NATURAL APPROVED and used for cosmetic and pharmaceutical use in the international market.

### 4.3. Cell Cultures

Human keratinocytes (HaCat) cell lines, purchased from AddexBio, catalog number T0020001, (AddexBio, San Diego, CA, USA) were cultured in DMEM high glucose medium supplemented with 10% Fetal Bovine Serum (FBS) and 1% penicillin/streptomycin.

HGFs were obtained from 10 donors, periodontally and systemically healthy, who underwent surgical third molar extraction after having signed the informed consent, according to the Italian legislation and to the Ethical Principles for Medical Research including Human Subjects of the World Medical Association (Declaration of Helsinki). The project received the approval by the Local Ethical Committee of the University of Chieti (Chieti, Italy; protocol number 1173, approved on 31 March 2016). After tooth extraction, gingiva fragments were immediately placed in DMEM, then rinsed in phosphate buffered saline buffer (PBS), cut into smaller pieces, and cultured in DMEM, with 10% FBS, 1% penicillin/streptomycin, and 1% fungizone (all purchased from EuroClone, Milan, Italy). After 10 days of culture, fungizone was removed and cells were used after 7–14 passages, as previously reported [30].

Undifferentiated human monocytes (CRL-9855™) were purchased from ATCC^®^ and sub-cultured in RPMI 1640 (Merck, Darmstadt, Germany) supplemented with 10% heat-inactivated fetal bovine serum (FBS), 1% penicillin/streptomycin, and 1% sodium pyruvate (all from Gibco, Invitrogen, Life Technologies, Carlsbad, CA, USA) at 37 °C and 5% CO_2_. For differentiation into macrophages, monocytes were seeded into 96 multi-well culture plates (0.5 × 10^4^ cells/well) and stimulated with 100 ng/mL of PMA (phorbol-12-myristate-13-acetate, purchased from Merck, Darmstadt, Germany, stock solution 1 mM in DMSO) in complete RPMI for 48 h at 37 °C and 5% CO_2_. All cell cultures were kept at 37 °C in a humidified atmosphere of 5% CO_2_.

### 4.4. Cell Treatments

HaCat were seeded into 96 multi-well culture plates at 8000 cells/well. After 24 h, cells were exposed to microfiltrated SS within DMEM culture medium at 1:20 (1.138 mg/mL), 1:40 (0.508 mg/mL), 1:60 (0.31 mg/mL), and 1:80 dilutions (0.247 mg/mL), for 24, 48, 72 h, and 6 days. The control sample (CTRL) was represented by HaCat treated with culture medium supplemented with sodium benzoate and potassium sorbate, at a percentage of 0.1%, to exclude preservative effects.

HGFs were seeded in 96 multi-well culture plates at 6700 cells/well. After 24 h, cells were firstly exposed to SS within DMEM culture medium at 1:40 (0.508 mg/mL), 1:60 (0.31 mg/mL), and 1:80 (0.247 mg/mL) dilutions for 24, 48, and 72 h, and the control sample was represented by HGFs exposed to culture medium supplemented with sodium benzoate and potassium sorbate, at a percentage of 0.1%, to exclude preservative effects (CTRL1).

Then HGFs were pre-treated with H_2_O_2_ 100 μM diluted in the culture medium for 3 h; following this, the medium containing H_2_O_2_ was discarded and replaced with a fresh one containing SS at 1:40 (0.508 mg/mL), 1:60 (0.31 mg/mL), and 1:80 (0.247 mg/mL) dilutions. The treatment was maintained for 6, 24, 48, and 72 h, according to the different biological essays. Control sample was represented by HGFs pre-treated with H_2_O_2_ 100 μM and then exposed to culture medium supplemented with sodium benzoate and potassium sorbate, at a percentage of 0.1%, to exclude preservative effects (CTRL2).

In the first set of experiments, macrophages were exposed to decreasing concentrations of SS, as were keratinocytes. In the second set of experiments, for establishing an inflamed environment, cells were stimulated with LPS 0.5 µg/mL (lipopolysaccharide from *E. coli*, purchased from Merck, Darmstadt, Germany, stock solution 1 mg/mL in water) and exposed to SS in parallel, as previously described.

### 4.5. MTT Assay (Cell Metabolic Activity)

Cell metabolic activity was measured by the MTT test (Merck Life Science, Milan, Italy), after 24, 48, 72 h, and 6 days of treatment for HaCat and after 24, 48, and 72 h for HGFs. At the established experimental times, the medium was discarded and replaced with a fresh one added with 0.5 mg/mL of MTT. After 5 h of incubation, the MTT solution was removed and DMSO was added for 30 min. The colored solution obtained by formazan crystal salt dissolution, formed through the capability of viable cells to reduce MTT into formazan, was read at 540 nm using a GO microplate spectrophotometer (Thermo Fisher Scientific, Waltham, MA, USA). The percentage of metabolically active cells was obtained through normalization with values of control cells (CTRL1 and CTRL2) (set as 100%).

### 4.6. MTS Assay (Cell Metabolic Activity)

The MTS test was performed in 96-well plates (Thermo Fisher Scientific, Waltham, MA, USA) as a measure of cell metabolic activity. At the established time point (24 and 48 h), the incubation medium was harvested for further analyses and a complete RPMI containing 3-(4,5-dimethylthiazol-2-yl)-5-(3-carboxymethoxyphenyl)-2-(4-sulfophenyl)-2*H*-tetrazolium (MTS) at a concentration of 0.5 mg/mL was added to each well. Cells were incubated for 4 h at 37 °C and 5% CO_2_. Following this, absorbance was measured at 490 nm using a spectrophotometer (Multiscan GO, Thermo Fisher Scientific, Waltham, MA, USA). The percentage of metabolically active cells in treated cultures was calculated by setting the untreated control to 100%.

### 4.7. LDH Assay (Cytotoxicity Assay)

To assess the membrane integrity of HGFs after SS treatment, the amount of lactate dehydrogenase (LDH) leakage into the culture medium was measured by means of CytoTox 96 non-radioactive cytotoxicity assay (Promega, Madison, WI, USA) following the manufacturer’s instructions. The assay was performed after 24 and 48 h of treatment and the measured LDH leakage was normalized with the MTT optical density values.

### 4.8. Haematoxylin/Eosin Staining

HGFs were seeded in sterile slide chambers (Millicell^®^ EZ slide, Merck Millipore, Burlington, MA, USA) at a density of 42,000 cells/well. After 24 h HGFs were washed twice with PBS with calcium and magnesium (EuroClone, Milan, Italy) and fixed with paraformaldehyde 4%. After 20 min, cells were washed again with PBS, permeabilized with Triton 0.15% in PBS for 3 min, and then stained with haematoxylin and eosin. Afterwards, slides were dehydrated through the ascending alcohols and mounted with a permanent xylene-based balsam. Computerized images were acquired by means of Leica DM 4000 light microscope (Leica Cambridge Ltd., Cambridge, UK) equipped with a Leica DFC 320 camera (Leica Cambridge Ltd., Cambridge, UK) and by using an image analysis software (LASX software, version 3.0 Leica Cambridge Ltd., Cambridge, UK).

### 4.9. Cell Cycle Analysis

HGFs were seeded in 6-well plates and treated for 6 h and 24 h, as previously described. For each time point, the cells were trypsinized, collected by centrifugation, and fixed overnight at 4 °C in cold ethanol 70% *v*/*v*. After fixation, cells were washed in cold phosphate buffer saline (PBS) and centrifuged at 4000 rpm for 10 min: the supernatants were discarded and each sample was stained with 300 µL of a staining solution containing PBS, RNase 100 µg/mL (stock solution 10 mg/mL in 10 mM sodium acetate buffer, pH 7.40), and propidium iodide (PI) 10 µg/mL (stock solution 1 mg/mL in water) (all purchased from Merck Life Science, Milan, Italy) and kept overnight at 4 °C in the dark. The PI fluorescence was detected by a flow cytometer equipped with a 488 nm laser (CytoFlex flow cytometer, Beckman Coulter, CA, USA) in the FL-3 channel. At least 10,000 events/sample were collected and analyzed with the CytExpert Software, version 2.3 (Beckman Coulter, CA, USA), and the percentages of cells in the G1, S, or G2 phase of the cell cycle were calculated using the ModFit LT™ Software, version 5.0 (Verity Software House, Topsham, ME, USA).

### 4.10. Protein Extraction and Western Blot Analysis

After 6 and 24 h of the treatments previously described, HGFs were trypsinized and collected by centrifugation at 1200× *g*. The pellets were resuspended in RIPA buffer with freshly added inhibitors (phenylmethylsulfonyl Fluoril (PMSF) 100 µg/mL, Aprotinin 10 µg/mL, Leupeptin 50 µg/mL and Sodium Orthovanadate 1 mM, all purchased from Merck Life Science, Milan, Italy), after rinses in ice-cold PBS. After cold centrifugation at 15,000× *g* for 15 min, the cell lysates were collected as the whole-cell fraction. The protein concentration was measured through a bicinchoninic acid assay (QuantiPro™ BCA Assay kit for 0.5–30 µg/mL protein, Merck Life Science, Milan, Italy) following the manufacturer’s instructions. For each sample, 20 µg of lysate were separated on a 10% (sodium dodecyl sulfate (SDS))-polyacrylamide gel by electrophoresis and transferred to a nitrocellulose membrane. The membrane was blocked in 5% non-fat milk, 10 mmol/L Tris-HCl pH 7.50, 100 mmol/L NaCl, 0.1% (*v*/*v*) Tween-20, and probed overnight at 4 °C under gentle shaking with the following primary antibodies: anti-phosphop44/42 MAPK (p-Erk1/2) rabbit monoclonal (1:1000), anti-p44/42 MAPK (Erk1/2) rabbit polyclonal (1:1000), anti-COX-2 rabbit monoclonal (1:1000), (all purchased from Cell Signaling Technology, Danvers, MA, USA), and anti-Actin mouse monoclonal (1:10,000) (Merck Life Science, Milan, Italy). Then, the membrane was incubated with specific IgG horseradish peroxidase (HRP)-conjugated secondary antibodies (Calbiochem, Darmstadt, Germany) and the immunoreactive bands were revealed using the ECL detection system (LiteAblot Extend Chemiluminescent Substrate, EuroClone S.p.a., Milan, Italy). Densitometric values of each band, expressed as integrated optical intensity, were estimated using a ChemiDoc™ XRS system and the QuantiOne 1-D analysis software, version 4.6.6 (BIORAD, Richmond, CA, USA). Values obtained were normalized based on the densitometric values of internal actin.

### 4.11. RNA Extraction

After 6 and 24 h of treatment, cells were trypsinized and collected by centrifugation. PureLink^®^ RNA Mini Kit (Life Technologies, Carlsbad, CA, USA) was used to extract the RNA according to the manufacturer’s instructions. An appropriate volume of supplied lysis buffer, supplemented with 1% of 2-mercaptoethanol, was added to each pellet followed by one volume of ethanol 70% *v*/*v*. Samples were then transferred into the spin cartridge for RNA extraction and purification. DNA contamination was removed by incubating samples with 80 µL of a DNase mixture (On-column PureLink^®^ Dnase Treatment, Life Technologies, Carlsbad, CA, USA) for 15 min. The extracted RNA was then eluted in 30 µL of nuclease-free water and its concentration (ng/µL) was determined through Qubit^®^ RNA BR Assay Kits (Life Technologies, Carlsbad, CA, USA), using a Qbit 4 Fluorometer reader (ThermoFisher Scientific Waltham, MA, USA).

### 4.12. Reverse Transcription (RT) and Real-Time RT-Polymerase Chain Reaction (Real-Time RT-PCR)

One µg of RNA was used to obtain the cDNA for each sample, using the high-capacity cDNA reverse transcription kit (Life Technologies, Carlsbad, CA, USA). Reactions were incubated in a thermal cycler at 25 °C for 10 min, 37 °C for 2 h, and 85 °C for 5 min. Gene expression was then determined by quantitative PCR using PowerUpTM SYBRTM Green Master Mix (2×) (Thermo Fisher Scientific, Waltham, MA, USA). Each amplification reaction was performed in a MicroAmp^®^ Optical 96-well reaction plate (Life Technologies, Carlsbad, CA, USA) in a reaction volume of 20 µL made up of 10 µL of SYBR Green, 1 µM of each primer (stock solution 10 µM), 10 ng of cDNA, and nuclease-free water. Primer sequences used are reported in Table 1.

The run method consists of the following steps: 50 °C for 2 min, 95 °C for 2 min, 40 cycles of amplification at 95 °C for 15 s, and 60 °C for 1 min, in QuantStudio 3 (Thermo Fisher Scientific, Waltham, MA, USA). QuantStudio™ Design and Analysis Software v1.5.1 (Thermo Fisher Scientific, Waltham, MA, USA) was used to elaborate gene expression data. The authenticity of the PCR products was verified by melt curve analysis. Each gene expression value was normalized to the expression level of glyceraldehyde 3-phosphate dehydrogenase (GAPDH). The fold changes of the investigated genes were expressed in relation to the level of GAPDH for each time point. The comparative 2^−∆∆Ct^ method was used to quantify the relative abundance of mRNA (relative quantification).

### 4.13. ELISA Analysis of Collagen I Secretion

After 24 and 48 h of HGF treatment, the amount of collagen type I secreted in the culture medium harvested from 6-well plates was detected using a human Col1 ELISA kit (cat. no. ACE-EC1-E105-EX, Cosmo Bio Co., Ltd., Tokyo, Japan). The coated wells supplemented by the kit were washed once with a wash buffer provided by the kit, then each sample was pipetted into the well working in duplicate. After incubation at room temperature for 1 h on a plate shaker, the blue conjugate and the yellow antibody were added. The plate was probed again for 2 h and wells were washed and incubated with the pNPP substrate for 45 min. Finally, a stop solution was added, and absorbance was read at 450 nm using a microplate reader (Multiskan GO, Thermo Fisher Scientific, Waltham, MA, USA). Col1 concentration, expressed as μg/mL, was calculated using a standard curve generated with standards provided by the kit using Prism 5.0 software (GraphPad, San Diego, CA, USA). Released Col1 amounts were normalized on viable cells optical density MTT values.

### 4.14. Statistical Analysis

Statistical analysis was performed using the GraphPad 7 software (GraphPad Software, San Diego, CA, USA) by means of ordinary one−way ANOVA followed by a post hoc Tukey’s multiple comparisons test.

## 5. Conclusions

In summary, this study focuses on an alternative SS extraction procedure able to guarantee the preservation of animal life, a lack of stress and/or dehydration for the snail, and the obtainment of a rich and beneficial product with very high physical and chemical properties. This innovative SS appears able to ensure promotion of neo angiogenesis, survival mechanisms, and new matrix deposition along with an appreciable anti-inflammatory effect. The time required to obtain the beneficial effects is often dependent on the health conditions of cells.

Due to the complexity and the several layers of the skin, these results represent a starting point for further biological evaluations of SS extracted with the Muller method. The study of new technological formulations, able to deliver an appreciable amount of slime to the deeper skin layers, along with the possibility to conjugate it with growth factors, antioxidants, antibiotics, etc., could represent the focus of future investigations.

## Figures and Tables

**Figure 1 molecules-28-01222-f001:**
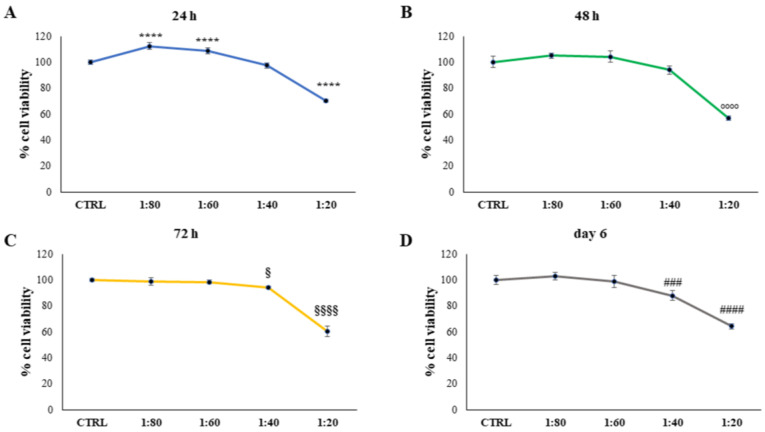
3-(4,5-dimethylthiazol-2-yl)-2,5-diphenyltetrazolium bromide (MTT) assay performed on HaCat cells treated for 24 h (**A**), 48 h (**B**), 72 h (**C**), and 6 days (**D**) with snail slime (SS) diluted to 1:80, 1:60, 1:40, and 1:20 within the culture medium. Metabolic activity was normalized to control cells treated with Dulbecco’s Modified Eagle Medium (DMEM). The most representative of three separate experiments is shown. Data are presented as the mean ± standard deviation. **** vs. control sample (CTRL) 24 h *p* < 0.0001; °°°° vs. CTRL 48 h *p* < 0.0001; §§§§ vs. CTRL 72 h *p* < 0.0001; § vs. CTRL 72 h *p* = 0.0127; #### vs. CTRL day 6 *p* < 0.0001; ### vs. CTRL day 6 *p* = 0.007.

**Figure 2 molecules-28-01222-f002:**
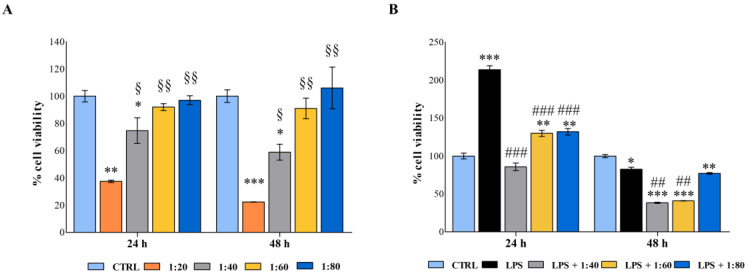
(**A**) 3-(4,5-dimethylthiazol-2-yl)-5-(3-carboxymethoxyphenyl)-2-(4-sulfophenyl)-2*H*-tetrazolium (MTS) assay performed on human macrophages treated for 24 and 48 h with SS diluted at 1:20, 1:40, 1:60, and 1:80 within the culture medium. (**B**) MTS assay performed on lipopolysaccharide (LPS)-stimulated human macrophages treated for 24 and 48 h with SS diluted at 1:40, 1:60, and 1:80 within the culture medium. Metabolic activity was normalized to control cells treated with RPMI culture medium. Data are presented as the mean ± standard deviation. * vs. CTRL *p* < 0.05, ** vs. CTRL *p* < 0.001, *** vs. CTRL *p* ≤ 0.0009; § vs. 1:20 *p* < 0.05, §§ vs. 1:20 *p* < 0.009, ## *p* < 0.01; ### *p* ≤ 0.006.

**Figure 3 molecules-28-01222-f003:**
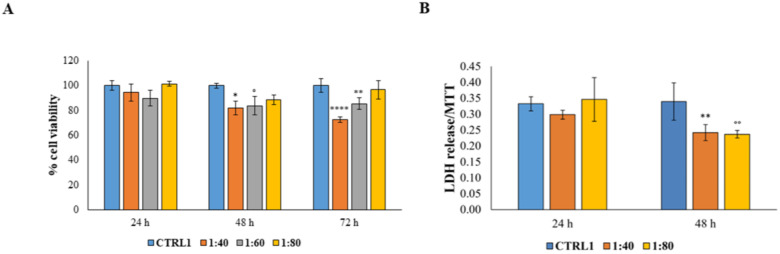
(**A**) MTT assay performed on HGFs treated for 24 h, 48 h, and 72 h with SS diluted at 1:40, 1:60, and 1:80 within the culture medium. Metabolic activity was normalized to control cells treated with DMEM culture medium (CTRL1). The most representative of three separate experiments is shown. Data are presented as the mean ± standard deviation. * vs. CTRL 48 h *p* = 0.0175; ° vs. CTRL 48 h *p* = 0.0307; **** vs. CTRL 72 h *p* < 0.0001; ** vs. CTRL 72 h *p* = 0.0096. (**B**) Cytotoxicity assay (LDH assay) of HGFs treated for 24 h and 48 h with SS diluted at 1:40 and 1:80 within the culture medium. The most representative of three separate experiments is shown. Data are presented as the mean ± standard deviation; LDH leakage is LDH release/MTT optical density (OD) values. ** vs. CTRL *p* = 0.0054; °° vs. CTRL *p* = 0.0038.

**Figure 4 molecules-28-01222-f004:**
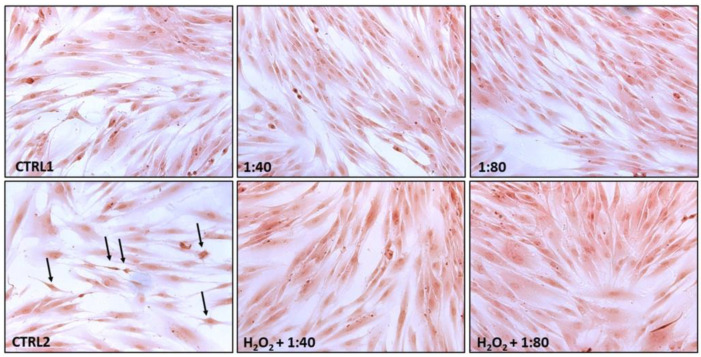
Hematoxylin/eosin staining of HGFs exposed for 24 h to DMEM culture medium (CTRL1), to SS diluted 1:40 and 1:80 (**upper** panel), and of HGFs pre-treated with H_2_O_2_ 100 µM for 3 h then exposed to DMEM culture medium (CTRL2) and to SS diluted at 1:40 and 1:80 (**lower** panel). Black arrows indicate suffering cells close to the detachment. Magnification 20×.

**Figure 5 molecules-28-01222-f005:**
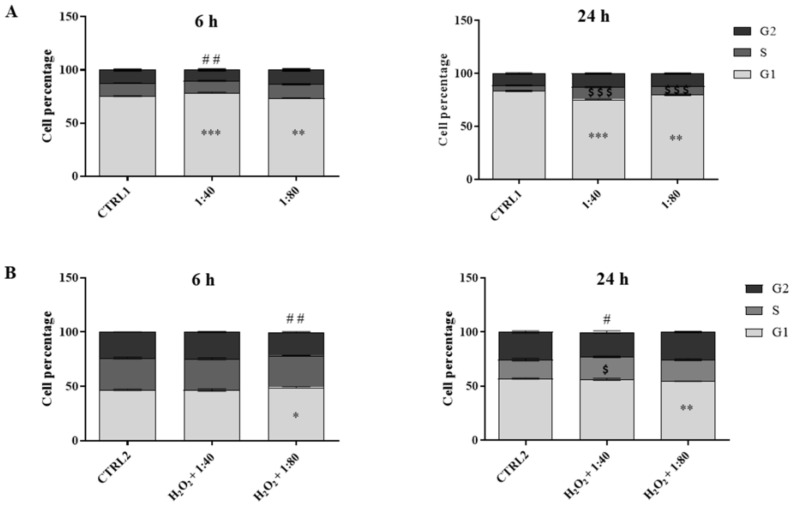
Cell cycle analysis in HGFs exposed for 6 and 24 h to DMEM culture medium (CTRL1), to SS diluted at 1:40 and 1:80 (panel (**A**)), and of HGFs pre-treated with H_2_O_2_ 100 µM then exposed to DMEM culture medium (CTRL2) and to SS diluted at 1:40 and 1:80 (panel (**B**)). The most representative of three separate experiments is shown. Data are presented as the mean ± standard deviation; the bar graph shows cell percentages in the cell cycle phases (G1, S, and G2) of HGFs. (**A**) G1 phase: *** vs. CTRL1 *p* < 0.0001; ** vs. CTRL1 *p* < 0.001. S phase: $$$ vs. CTRL1 *p* < 0.0001. G2 phase: ## vs. CTRL1 *p* < 0.0002. (**B**) G1 phase: ** vs. CTRL2 *p* < 0.0072; * vs. CTRL2 *p* < 0.014. S phase: $ vs. CTRL2 *p* < 0.017. G2 phase: ## vs. CTRL2 *p* < 0.0005; # vs. CTRL2 *p* < 0.019.

**Figure 6 molecules-28-01222-f006:**
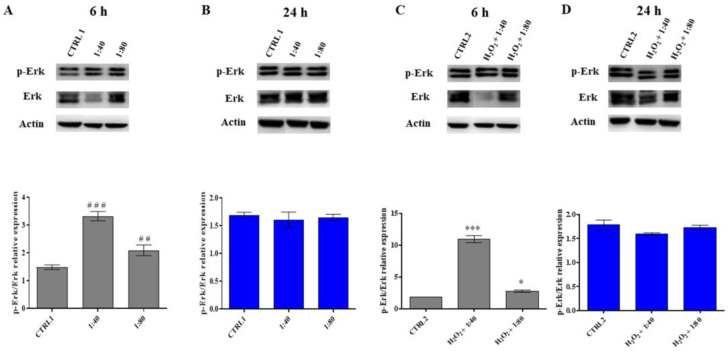
p44/42 MAPK (Erk) and phospho-p44/42 MAPK (p-Erk) expression levels in HGFs exposed to DMEM culture medium (CTRL1), to SS diluted 1:40 and 1:80 for 6 and 24 h (panels (**A**,**B**)) and of HGFs pre-treated with H_2_O_2_ 100 µM then exposed to DMEM culture medium (CTRL2) and to SS diluted 1:40 and 1:80 (panels (**C**,**D**)). The most representative of three separate experiments is shown. Actin is used as a loading control. The bar graph displays densitometric values expressed as mean ± standard deviation of p-Erk/Erk ratio normalized on loading control. (**A**) vs. CTRL1 ### *p* < 0.0001, ## *p* < 0.001; (**C**) vs. CTRL2 *** *p* < 0.0001, * *p* < 0.01.

**Figure 7 molecules-28-01222-f007:**
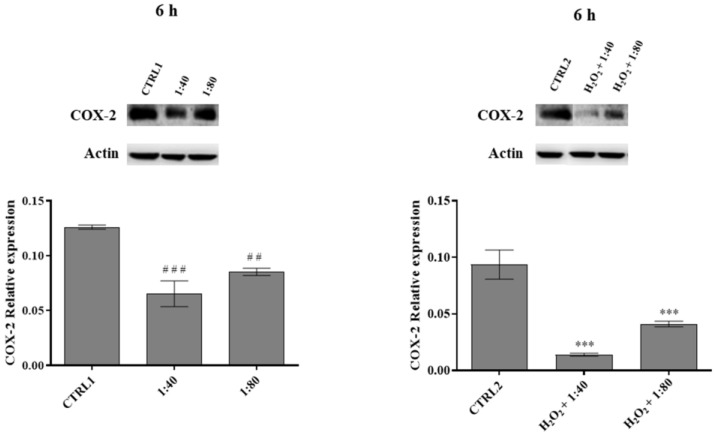
Cyclooxygenase-2 (COX-2) expression levels in HGFs exposed to DMEM culture medium (CTRL1), to SS diluted 1:40 and 1:80 (**left** histogram), and of HGFs pre-treated with H_2_O_2_ 100 µM then exposed to DMEM culture medium (CTRL2) and to SS diluted 1:40 and 1:80 (**right** histogram) for 6 h. The most representative of three separate experiments is shown. Actin is used as a loading control. The bar graph displays densitometric values expressed as mean ± standard deviation of COX-2 normalized on loading control. Left histogramvs. CTRL1 ### *p* < 0.0001, ## *p* < 0.001; right histogram vs. CTRL2 *** *p* < 0.0001.

**Figure 8 molecules-28-01222-f008:**
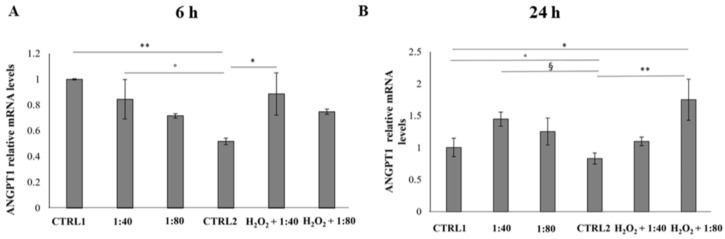
Relative gene expression of Angiopoietin 1 (ANGPT1) in HGFs exposed to DMEM culture medium (CTRL1), SS diluted 1:40 and 1:80, HGFs pre-treated with H_2_O_2_ 100 µM then exposed to DMEM culture medium (CTRL2) and to SS diluted 1:40 and 1:80 for 6 h (**A**) and 24 h (**B**). Data are expressed as relative to CTRL1 (calibrator sample defined as 1). The most representative of three separate experiments is shown. The bar graph displays densitometric values expressed as mean ± standard deviation, Y-axis, fold change. 6 h: ** CTRL1 vs. CTRL2 *p* = 0.0073; * CTRL2 vs. H_2_O_2_ + 1:40 *p* = 0.0257; ° CTRL2 vs. 1:40 *p* = 0.0427 24 h: ** CTRL2 vs. H_2_O_2_ + 1:80; * CTRL1 vs. H_2_O_2_ + 1:80 *p* = 0.0207; ° CTRL1 vs. CTRL2 *p* = 0.0635; § CTRL2 vs. 1:40 *p* = 0.0474.

**Figure 9 molecules-28-01222-f009:**
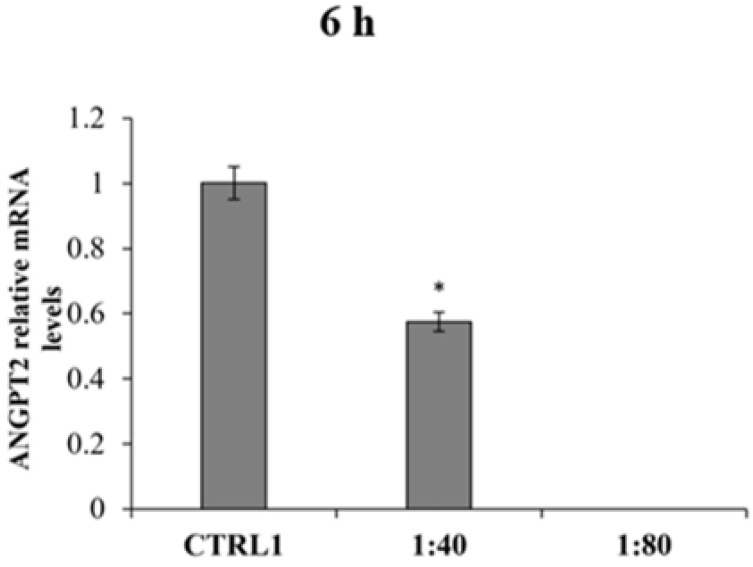
Relative gene expression of Angiopoietin 2 (ANGPT2) in HGFs exposed to DMEM culture medium (CTRL1), to SS diluted 1:40 and 1:80 for 6 h. Data are expressed as relative to CTRL1 (calibrator sample, defined as 1). The most representative of three separate experiments is shown. The bar graph displays densitometric values expressed as mean ± standard deviation, Y-axis, fold change. vs. CTRL1 * *p* = 0.0135.

**Figure 10 molecules-28-01222-f010:**
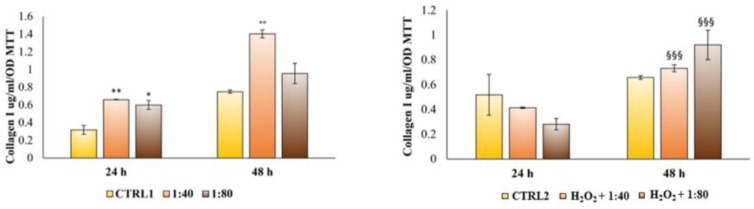
Enzyme-linked immunosorbent assay (ELISA) for collagen I secretion in HGFs exposed to DMEM culture medium (CTRL1), to SS diluted 1:40 and 1:80 and of HGFs pre-treated with H_2_O_2_ 100 µM then exposed to DMEM culture medium (CTRL2) and to SS diluted 1:40 and 1:80 for 24 and 48 h. Secretion levels are reported as μg/mL/MTT OD viable cells. The most representative of three separate experiments is shown. The bar graph displays densitometric values expressed as mean ± standard deviation. ** vs. CTRL1 *p* = 0.0058, * vs. CTRL1 *p* = 0.0102, °° vs. CTRL1 *p* = 0.0045, §§§ vs. CTRL2 *p* = 0.0003.

**Table 1 molecules-28-01222-t001:** Primer sequences for qPCR.

Gene	Sequence (5′ to 3′)
GAPDH-FW	GGGTGTGAACCATGAGAAGTA
GAPDH-RW	ACTGTGGTCATGAGTCCTTC
ANGPT1-FW	AGGTGTTTTACTAAAGGGAGGAA
ANGPT1-RW	AACCTCCCCCATTGACATCC
ANGPT2-FW	ACCTGTTGAACCAAACAGCG
ANGPT2-RW	GTCGAGAGGGAGTGTTCCAAG

## Data Availability

No new data were created or analyzed in this study. Data sharing is not applicable to this article.

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
