# Peer review of "Snail Slime Extracted by a Cruelty Free Method Preserves Viability and Controls Inflammation Occurrence: A Focus on Fibroblasts"

_molecules, 2023, doi:10.3390/molecules28031222_

Round 1

Reviewer 1 Report

The manuscript from Ricci et al. is an in vitro study investigating the toxicity, anti-inflammatory and antiaging potential and the mechanism of action of snail slime obtained by a new extracting cruelty-free procedure. These bioactive properties and the cellular mechanisms thereof were investigated by in vitro cellular tests using human skin cells, keratinocytes, fibroblasts and macrophages since these are cells coming to contact with snail slime cosmetical preparations.

There are too many criticisms in this paper that should be addressed by the authors before considering a resubmission.

Here is the list of major points:

The first point is, I understand that the Muller slime extraction methodology is patented, but in the literature no papers on the composition of such slime extract are already available, so the authors should perform a composition analysis of the extract, to confirm the retrieval and the yield of the typical compounds contained in the snail slime after the procedure of extraction. Furthermore, just using scalar dilutions without indicating at least a protein content or the dry weight content of the dilutions used, does not allow to understand the quantity of bioactive molecules tested throughout the paper.

For what concerns the HGF data: 1) the authors should try to explain the inconsistency between the slight cytotoxic effect at 48h in the MTT test, and a reduction of cytotoxicity at the same time deducible from the LDH assays. These experiments seem to show opposite effects. 2) Since there is no real cytotoxic effect of the H2O2 treatment (MTT test) and therefore there is not a real testing of the rescuing capabilities of the SS on stressed/dying cells, maybe a higher H2O2 concentration should have been tried to test the SS ability to increase cell survival. Furthermore, in the histological/morphological analysis of H2O2 treated or untreated cells, the fact that a lower cell confluence is observed in the slides with H2O2 treatment should correspond to lower MTT values, however this is not observable in your experiments. 3) Also, the cell cycle analysis is misleading: these data are in contrast with the MTT tests which show no cell growth in the H2O2-treated cells, although here the increase of S-G2 cells seems to show a proliferative effect of H2O2 by itself and enhanced by the SS treatment, as compared to untreated control cells. 4) Can the author explain why there is a higher expression of COX-2 already in control cells (CTRL1) since this protein is inducible and usually not expressed, or really low expressed in resting cells. Its expression should increase only after an inflammatory stimulus. Are the gingival fibroblasts somehow already activated by some procedure during the isolation protocol?

For what concerns the macrophages experiments, the authors did not test a real an anti-inflammatory effect on monocyte/macrophages but just the proliferation under LPS treatment. To declare an anti-inflammatory effect there should be an investigation of the SS on immune cells themselves, like a gene expression profile of inflammatory factors, cytokines, after LPS treatment in presence or absence of the SS, prostaglandins quantification in the medium and so on.

Regarding the gene expression analysis, it would be important to show the relative expression of H2O2 treated cells as respect to control cells (CTRL1) to understand the behaviour of the stressing stimulus by itself on the gene expression of the two investigated factors, in this graph it is not possible to understand the effect of H2O2 on the cells as compared to the untreated cells: is the same, increases or diminishes? and then what is the contribute of SS? The authors should show all the data compared to the CTRL1 at 6 h and at 24h. Furthermore, the data on the Angpt2 are not clear, the expression was completely abolished by the SS at the highest dilution and by H2O2 treatment? Normally there is a decrease of gene expression that is still measurable, there seems to be some technical problem here.

Author Response

- The first point is, I understand that the Muller slime extraction methodology is patented, but in the literature no papers on the composition of such slime extract are already available, so the authors should perform a composition analysis of the extract, to confirm the retrieval and the yield of the typical compounds contained in the snail slime after the procedure of extraction. Furthermore, just using scalar dilutions without indicating at least a protein content or the dry weight content of the dilutions used, does not allow to understand the quantity of bioactive molecules tested throughout the paper.

Answer: We thank the referee for this right consideration. The Technical Data Sheet, reporting all the required information (organoleptic, chemical, physical, microbiological characteristics, determination of heavy metals and allergens), has been added to the paper as supplementary material. Regarding the use of SS scalar dilutions, unfortunately, when the researchers work with extracts which have not a molecular weight, they are forced to administer them in percentages, if solids, ore in dilutions if they are liquid.   

- For what concerns the HGF data:

1) the authors should try to explain the inconsistency between the slight cytotoxic effect at 48h in the MTT test, and a reduction of cytotoxicity at the same time deducible from the LDH assays. These experiments seem to show opposite effects.

Answer: We thank the referee for its valuable observation, probably the authors did not clearly explain the biological aspects measured by MTT and LDH test. There is not a contradiction between the two data as they provide information regarding different biological aspects, thus they don’t necessarily have to disclose the same trend. MTT test, in fact, measures the percentage of metabolically active cells, thus providing an information regarding the percentage of viable cells. Conversely, LDH test evaluates the release, within the culture medium, of lactate dehydrogenase upon cell lysis, thus providing an information regarding necrotic cells (definitely dead cells) which are not metabolically active. The data shown in figure 3 suggest that after 48 of treatment there is a slight decrease of HGFs treated with SS at 1:40 and 1:60 dilutions viability respect to untreated cells, but SS, also at the higher concentration (1:40) can’t be considered cytotoxic. Thus, LDH result definitely excludes SS toxicity showing that it is perfectly tolerated by HGFs.

2) Since there is no real cytotoxic effect of the H2O2 treatment (MTT test) and therefore there is not a real testing of the rescuing capabilities of the SS on stressed/dying cells, maybe a higher H2O2 concentration should have been tried to test the SS ability to increase cell survival. Furthermore, in the histological/morphological analysis of H2O2 treated or untreated cells, the fact that a lower cell confluence is observed in the slides with H2O2 treatment should correspond to lower MTT values, however this is not observable in your experiments.

Answer: H2O2 concentration (100 µM for 3h) has been chosen after a screening experiment in which H2O2 concentrations, ranging from 25 to 800 µM, were tested. Considering that the aim of our experimental model was to provide a stimulus able to trigger an oxidative stress without affecting cell viability and, according to the screening experiment (performed by MTT), 100 µM was chosen as final concentration as it was able to keep a high level of cell viability. Conversely, a higher and thus toxic, H2O2 concentration would have provoked atoo high rate of dead cells excluding the possibility to investigate both the SS rescuing capability and the molecular mechanisms underlying its biologic effect.  

Regarding the morphological analysis shown in figure 5, it refers to a 24 h treatment, and, as the referee can see, the MTT data of 24 h evidences, even if there is not a statistical significance, a slight increase in cell viability in HGFs treated with SS, especially if administered at 1:40 dilution, which is the same situation shown by the morphological analysis.

3) Also, the cell cycle analysis is misleading: these data are in contrast with the MTT tests which show no cell growth in the H2O2-treated cells, although here the increase of S-G2 cells seems to show a proliferative effect of H2O2 by itself and enhanced by the SS treatment, as compared to untreated control cells.

Answer: The authors think that it is not possible to compare the MTT results with the results of the cell cycle analysis for different reasons.

Firstly, once again, the two analyses provide information regarding different biological cell processes. As already explained, MTT measures the percentage of metabolically active cells. Thus, thus the viability rate is an indirect information obtained from the number of metabolically active cells, as a consequence, this assay can’t be used to determine how much the cells are proliferating. In addition, in the MTT results, the two controls (CTRL1 and CTRL2) are both set at 100% by the operator showing relative measurements (Figs 3A and 4), thus, it is not possible to compare the number of viable cells in the two experimental points. The cell cycle analysis instead is a very accurate measurement that provides information regarding the distribution of viable cells in the three phases of the cell cycle thus giving information regarding the cell proliferation.

Secondly, unlike the MTT assay, the cell cycle analysis considers only live cells: in fact, when cells are collected for the assay, only adherent cells (live cells) are harvested thus excluding detached cells (dead cells).

However, the higher proliferation recorded in CTRL2 compared to CTRL1 can be explained admitting that when cells (such as tenocytes, fibroblasts and others) are exposed to an inflammatory stimulus, three main phases allow the healing process. The first phase is a reparative process, mainly characterized by a high proliferation rate of cells, an automatic and compensatory response to a stressed situation. The proliferative step is then followed by the remodeling and the consolidation step, as reported by Docheva D et al. [Adv Drug Deliv Rev. 2015 Apr;84:222-39]. In CTRL1, primary fibroblasts are in a physiological healthy condition and the proliferation proceeds slowly, in accordance with our cell cycle result. The H2O2-stimulus in CTRL2 triggers an automatic response in the cells, which react by increasing their proliferation, as evidenced by the cell cycle results.

The authors did not underline and explained this aspect in the discussion as the paper is focused on explanations regarding the biological effect of SS, however, if the referee considers worthwhile to shed light on this point, a paragraph can be added in the discussion section.

4) Can the author explain why there is a higher expression of COX-2 already in control cells (CTRL1) since this protein is inducible and usually not expressed, or really low expressed in resting cells. Its expression should increase only after an inflammatory stimulus. Are the gingival fibroblasts somehow already activated by some procedure during the isolation protocol?

Answer: Thank you for this observation. The gingiva biopsy from which HGFs were isolated was extracted closely to a surgical site, thus, it could be likely that these cells, having undergone the stressful situation of the surgical intervention, could reveal a high basal level of COX-2. This hypothesis is also confirmed by the fact that in some previous papers, we already found a high release and/or expression of pro-inflammatory markers such as PGE2 [Di Nisio C et al., Int Endod J. 2013 May;46(5):466-76].

- For what concerns the macrophages experiments, the authors did not test a real an anti-inflammatory effect on monocyte/macrophages but just the proliferation under LPS treatment. To declare an anti-inflammatory effect there should be an investigation of the SS on immune cells themselves, like a gene expression profile of inflammatory factors, cytokines, after LPS treatment in presence or absence of the SS, prostaglandins quantification in the medium and so on.

Answer:  The authors agree with the referee and are aware that to demonstrate a counteraction of inflammation in immunocompetent cells additional and in-depth experiments, correctly mentioned in your comment, are required. The counteraction of macrophage activation by the SS and the re-configuration of their augmented metabolism under LPS stimulation is only a starting point and do not demonstrate an anti-inflammatory effect in these cells. As mentioned within the introduction and the discussion sections, the first aim of the authors was to preliminary screen the SS in the three main cellular skin components (keratinocytes, macrophages and fibroblasts). Next, we have investigated the SS molecular effects only on one cell type (fibroblasts). In the light of your observation, the authors decided to slightly modify the title, highlighting that our paper focuses on cell responses towards inflammation mainly in fibroblasts. Additionally, the discussion about data referred to macrophages has been modified as well. More in depth analyses are ongoing on macrophages, to demonstrate a mechanism of action induced by the SS in these cells.

- Regarding the gene expression analysis, it would be important to show the relative expression of H2O2 treated cells as respect to control cells (CTRL1) to understand the behaviour of the stressing stimulus by itself on the gene expression of the two investigated factors, in this graph it is not possible to understand the effect of H2O2 on the cells as compared to the untreated cells: is the same, increases or diminishes? and then what is the contribute of SS? The authors should show all the data compared to the CTRL1 at 6 h and at 24h. Furthermore, the data on the Angpt2 are not clear, the expression was completely abolished by the SS at the highest dilution and by H2O2 treatment? Normally there is a decrease of gene expression that is still measurable, there seems to be some technical problem here.

Answer: We repeated the gene expression data processing choosing CTRL1 as calibrator sample, as requested by the referee.

Firstly, the new obtained histograms evidence the effect of H2O2 treatment on the cells, compared to the untreated cells underlining that, after both 6 and 24 h, the exposure to H2O2 significantly decreases ANGPT1 gene expression.

Secondly, the new graphs highlight the positive SS contribute to ANGPT1 gene expression. In fact, after 6h of treatment, SS administration is definitely responsible for an appreciable increase of ANGPT1 gene expression respect to cells receiving the H2O2 pro-inflammatory stimulus (CTRL2 vs H2O2+1:40): its level appears approximately re-established, by SS treatment, at basal level. The effect is also confirmed after 24h of treatment as, a valuable increase in ANGPT1 gene expression is evidenced when cells are exposed to H2O2 first and then are treated with SS 1:80 (CTRL2 vs H2O2+1:80).

Based on this new data processing, a new description of ANGPT1 gene expression results and some modifications in the discussion section, have been added, eliminating the previous ones.

Regarding ANGPT2 gene expression results, we apologize because, probably, our explanation within the text was not clear. The ANGPT2 gene expression at the highest SS dilution and after the H2O2 treatment was totally absent, this means that ANGPT2 gene is not expressed at all. This is not the consequence of a technical problem as, a gene can be scarcely expressed and in this case its expression can be measurable, or, can be totally not expressed, and, in this case, not measurable. In the paper reported below, for example, in a similar way, ANGPT2 gene expression was not recorded in some experimental points [Ricci A et al. Molecules. 2022 Jan 31;27(3):957]. If the referee wants, we can send the file with the raw data produced by the Real Time instrument to check the ANGPT2 lack of expression in most of experimental points.

Grammar and English language have been revised and corrected throughout the manuscript.

Reviewer 2 Report

In the manuscript entitled “Snail slime extracted by a cruelty free method preserves fibroblasts viability and controls inflammation occurrence” the authors presented cruelty free method of snail slime extraction  promotes neo angiogenesis, survival mechanisms and new matrix deposition along with an appreciable anti-inflammatory effect.

This manuscript makes contribution to the field and merits for publication.

Author Response

In the manuscript entitled “Snail slime extracted by a cruelty free method preserves fibroblasts viability and controls inflammation occurrence” the authors presented cruelty free method of snail slime extraction promotes neo angiogenesis, survival mechanisms and new matrix deposition along with an appreciable anti-inflammatory effect.

This manuscript makes contribution to the field and merits for publication.

Answer: The authors thank the referee for his positive feedback and for his recommendation for publication.

Reviewer 3 Report

The manuscript entitled "Snail slime extracted by a cruelty free method preserves fibroblasts viability and controls inflammation occurrence" is a hypothesis driven work submitted by the authors. Authors have made extensive investigations on the mechanism of antiinflammatory nature and its application using different experimental models. I feel some areas of the manuscript still need improvement; the comments on the work has been incldued below.

The abstract look oversimplified; authors need to organize the section with a strong background information followed by precise objectives and methodology,  a short decription on major finding with quantitative data and conclusion.

Authors focused on the antiinflammatory aspects of SS; hence, there need an extensive outline on the inflammation mechanisms and molecular events in skin pathology. 

How the authors identified the species of animal? Any taxonomic expertise involved in it?

What about the composition of the slime from the species? Authors need to include data on the characterization aspects of the slime, since it is a chemistry journal. I think HPLC methods of characterization of mucopolysaccharides are available.

The figure 1 can be represented as a line graph, especially when indicating a cell viability aspect. I suggest to start with control, 1:80, 1: 60, 1:40, and 1:20 dilution.

Line 112: "significantly decreases the percentages of cell metabolic cells with" check the accuracy 

What was the cell seeding density for MTT assay, because authors could maintain for 6 days without significant damage. I think it is a long period, in a cytotoxicity analysis.

In addition, Figure 1 shows a higher cell viability (cell growth or metabolic increase) in low dose (1:80 or 1:60) treatments. Explain the possible reason for the observed increase in viability

Authors used LPS as stimulant of inflammation. However, they neglected the LPS activation mechanism in discussion. In addition, the LPS-mediated toll like receptor activation (and its possible future implications in various diseases) needs to be discussed. Authors may see the following articles

https://doi.org/10.2174%2F1871526510808030144

https://doi.org/10.2174/1389450120666190222181506

https://doi.org/10.17179/excli2020-3114

https://doi.org/10.3390/ijms21228560

In Figure 11 and some other places, authors missed to make the 2 in H2O2 as subscript

Figures 9 and 10 may follow the color pattern as in figure 8 (i think the grayscale is more presentable)

Author Response

- The abstract look oversimplified; authors need to organize the section with a strong background information followed by precise objectives and methodology, a short description on major finding with quantitative data and conclusion.

Answer: Probably the referee is right when he states that the abstract is too short. However, the instruction for authors of the journal indicate that the number of abstract words can’t exceed 200. The methodology used in the experiments were already indicated (…cell viability through MTT test, cytotoxicity by LDH assay, morphology by haematoxylin-eosin staining, gene and protein expression through Real-Time PCR and western blot, cell cycle phases by flow cytometry and collagen secretion trough ELISA test, were measured…). By exceeding of 20 the limit of 200 words, to better clarify the background, a sentence has been added.

- Authors focused on the antiinflammatory aspects of SS; hence, there need an extensive outline on the inflammation mechanisms and molecular events in skin pathology. 

Answer: The authors agree with the referee and they reported more precisely the involvement of inflammation in skin pathology in the discussion section.

- How the authors identified the species of animal? Any taxonomic expertise involved in it?

Answer: The International Institution of Heliciculture, responsible for the study of snails and the extraction of SS in the last 30 years, based on the scientific literature and on the morphology of the snails, is capable to recognize snails of Helix Aspersa species (Aspersa Muller, Aspersa Maxima and Aspersa Aspersa).

- What about the composition of the slime from the species? Authors need to include data on the characterization aspects of the slime, since it is a chemistry journal. I think HPLC methods of characterization of mucopolysaccharides are available.

Answer: Thank you for this observation. We definitely agree with the referee request of more technical data. As a consequence, the methods used to analyse and characterize the quality of the SS, obtained from the extractor machine, are reported in detail within the table provided in the supplementary material.

- The figure 1 can be represented as a line graph, especially when indicating a cell viability aspect. I suggest to start with control, 1:80, 1: 60, 1:40, and 1:20 dilution.

Answer: As suggested by the referee the histogram in figure 1 has been replaced by line graphs. We prepared a line graph for each experimental time to better visualize the viability differences. However, if the referee prefers, we can also collapse all the four lines in a single line graph.

- Line 112: "significantly decreases the percentages of cell metabolic cells with" check the accuracy 

Answer:  Thank you for this observation, we have modified the text and now the sentence is correct. “After 48 h, the co-treatment LPS/SS significantly decreases the percentages of metabolically active cells with respect to CTRL and LPS”

- What was the cell seeding density for MTT assay, because authors could maintain for 6 days without significant damage. I think it is a long period, in a cytotoxicity analysis.

Answer: The authors thank the referee for its appreciable question and apologize for the lack of information regarding the cell seeding density. The information has been added. Moreover, regarding the 6 days period of treatment, the possibility of a cell damage can be excluded as HaCat are healthy cells thus provided of contact inhibition when they reach a full confluence state. However, we agree with the referee when he states that it is a very long period, in fact, as the referees could see, this experimental time was removed from next experiments.

- In addition, Figure 1 shows a higher cell viability (cell growth or metabolic increase) in low dose (1:80 or 1:60) treatments. Explain the possible reason for the observed increase in viability

Answer: as specified in the second paragraph of the discussion section, the significant increase in keratinocyes viability when SS is administered at 1:60 and 1:80 is probably due to the large amount of glycolic acid in SS. Glycolic acid possesses, as mentioned, a well known keratinolytic effect, thus, a lower concentration of SS, and, as a consequence, of glycolic acid, is better tolerated by keratinocytes.

- Authors used LPS as stimulant of inflammation. However, they neglected the LPS activation mechanism in discussion. In addition, the LPS-mediated toll like receptor activation (and its possible future implications in various diseases) needs to be discussed. Authors may see the following articles

https://doi.org/10.2174%2F1871526510808030144

https://doi.org/10.2174/1389450120666190222181506

https://doi.org/10.17179/excli2020-3114

https://doi.org/10.3390/ijms21228560

Answer: The authors are thankful for this observation. The discussion section has been improved accordingly.

- In Figure 11 and some other places, authors missed to make the 2 in H2O2 as subscript

Answer: The legends of figures 4, 9 and 11 have been corrected, as requested by the referee.

- Figures 9 and 10 may follow the color pattern as in figure 8 (i think the grayscale is more presentable)

Answer: the color pattern of figures 9 and 10 has been changed as suggested by the referee.

Grammar and English language have been revised and corrected throughout the manuscript.

Round 2

Reviewer 1 Report

There are still two important points not sufficiently addressed:

1)

Authors' response to reviewer question round 1:

Answer: We thank the referee for this right consideration. The Technical Data Sheet, reporting all the required information (organoleptic, chemical, physical, microbiological characteristics, determination of heavy metals and allergens), has been added to the paper as supplementary material. Regarding the use of SS scalar dilutions, unfortunately, when the researchers work with extracts which have not a molecular weight, they are forced to administer them in percentages, if solids, ore in dilutions if they are liquid.   

 Reviewer’s reply:

I appreciate the addition of the technical data sheet that allows to understand better the composition of the extract, but I disagree on the statement that when you work with heterogeneous extracts which, of course, do not rely on a single molecular weight, it’s not possible to quantify the material contained in the extract that ultimately allows the researchers to make comparisons with the work of others. The authors will find thousands of papers in which, when dealing with extracts from the natural world, the quantification of the raw material used is always reported. The authors must express the dilutions used, if not by titration of a major component of the extract, at least as the dry weight (mg or mg)/ml of extract which can be easily obtained by a freeze drying process of a known aliquot of the raw, liquid extract which can then be weighted.

2)

Authors' response to reviewer question round 1:

Answer: H2O2 concentration (100 µM for 3h) has been chosen after a screening experiment in which H2O2 concentrations, ranging from 25 to 800 µM, were tested. Considering that the aim of our experimental model was to provide a stimulus able to trigger an oxidative stress without affecting cell viability and, according to the screening experiment (performed by MTT), 100 µM was chosen as final concentration as it was able to keep a high level of cell viability. Conversely, a higher and thus toxic, H2O2 concentration would have provoked atoo high rate of dead cells excluding the possibility to investigate both the SS rescuing capability and the molecular mechanisms underlying its biologic effect.  

Regarding the morphological analysis shown in figure 5, it refers to a 24 h treatment, and, as the referee can see, the MTT data of 24 h evidences, even if there is not a statistical significance, a slight increase in cell viability in HGFs treated with SS, especially if administered at 1:40 dilution, which is the same situation shown by the morphological analysis.

 Reviewer’s reply:

The explanation by the authors does not answer to the criticism raised by the reviewer. The H2O2 treatment does not show any effect detectable by the MTT test, thus the use of the SS extract does not counteract anything that is measurable through the MTT test, which is indeed a test aiming to highlight the cytotoxicity of a given treatment. If the authors want to measure a rescuing activity of the extract by using the MTT test as a means to detect this activity, the positive control, i.e. the H2O2-treated cells, must show a cytotoxic effect in order to measure the eventual beneficial activity of the extract in counteracting said cytotoxicity. Conversely, if they claim that they wanted to measure the rescuing as an antioxidant activity of the extract upon the H2O2-treatment of cells, they should have performed a dedicated experiment measuring for example the impact on ROS production or the intracellular GSH/GSSG ratio or the intracellular activitiy of some antioxidant enzyme and so on. In this form figure 4 does not give any useful information, if the authors don’t want to show the effects of the SS extract on higher concentrations of H2O2 causing a measurable cytotoxic activity, they should remove the figure, at least in this way the discrepancy with the images of figure 5 will disappear.

Author Response

I appreciate the addition of the technical data sheet that allows to understand better the composition of the extract, but I disagree on the statement that when you work with heterogeneous extracts which, of course, do not rely on a single molecular weight, it’s not possible to quantify the material contained in the extract that ultimately allows the researchers to make comparisons with the work of others. The authors will find thousands of papers in which, when dealing with extracts from the natural world, the quantification of the raw material used is always reported. The authors must express the dilutions used, if not by titration of a major component of the extract, at least as the dry weight (mg or mg)/ml of extract which can be easily obtained by a freeze drying process of a known aliquot of the raw, liquid extract which can then be weighted.

Answer: the quantification of the raw material used for the experiments has been carried out by freeze drying and then expressed in the materials and methods section as mg/ml, as requested by the referee.

The explanation by the authors does not answer to the criticism raised by the reviewer. The H2O2 treatment does not show any effect detectable by the MTT test, thus the use of the SS extract does not counteract anything that is measurable through the MTT test, which is indeed a test aiming to highlight the cytotoxicity of a given treatment. If the authors want to measure a rescuing activity of the extract by using the MTT test as a means to detect this activity, the positive control, i.e. the H2O2-treated cells, must show a cytotoxic effect in order to measure the eventual beneficial activity of the extract in counteracting said cytotoxicity. Conversely, if they claim that they wanted to measure the rescuing as an antioxidant activity of the extract upon the H2O2-treatment of cells, they should have performed a dedicated experiment measuring for example the impact on ROS production or the intracellular GSH/GSSG ratio or the intracellular activity of some antioxidant enzyme and so on. In this form figure 4 does not give any useful information, if the authors don’t want to show the effects of the SS extract on higher concentrations of H2O2 causing a measurable cytotoxic activity, they should remove the figure, at least in this way the discrepancy with the images of figure 5 will disappear.

Answer: considering that: (i) the concentration of H2O2, as mentioned in the previous round, can’t be increased because the basis of molecular investigations is to apply subtoxic concentrations of the stimulus to preserve a high rate of cell viability aiming at investigating the recruitment of molecular pathways; (ii) the situation shown in the morphological analysis, in our opinion, does not show a discrepancy with the MTT results after 24 h as both the two data evidence a slight reduction of H2O2-exposed cells viability; (iii) the SS rescuing capability has been always related to appropriate parameters such as Erk expression (survival protein), S phase of the cell cycle etc, without taking into consideration the MTT results as the authors perfectly know that this statement requires specific and detailed results which can’t be MTT results; the authors do not totally agree with the option of deleting figure 4 from the manuscript, however, as the referee considers it necessary, figure 4 and its results description have been removed from the paper. 

Reviewer 3 Report

The article has been improved based on the comments

Author Response

The authors thank the referee for his positive feedback.